# Effectiveness of FLASH vs. Conventional Dose Rate Radiotherapy in a Model of Orthotopic, Murine Breast Cancer

**DOI:** 10.3390/cancers17071095

**Published:** 2025-03-25

**Authors:** Stavros Melemenidis, Vignesh Viswanathan, Suparna Dutt, Naviya Kapadia, Brianna Lau, Luis A. Soto, M. Ramish Ashraf, Banita Thakur, Adel Z. I. Mutahar, Lawrie B. Skinner, Amy S. Yu, Murat Surucu, Kerriann M. Casey, Erinn B. Rankin, Kathleen C. Horst, Edward E. Graves, Billy W. Loo, Frederick M. Dirbas

**Affiliations:** 1Department of Radiation Oncology, Stanford University School of Medicine, Stanford, CA 94305, USA; stavmel@stanford.edu (S.M.); vignesh1984@gmail.com (V.V.); sdutt@stanford.edu (S.D.); brianna.lau@my.rfums.org (B.L.); luisasr@stanford.edu (L.A.S.); ramish.ashraf@stanford.edu (M.R.A.); lawrie.skinner@stanford.edu (L.B.S.); amysyu@stanford.edu (A.S.Y.); surucu@stanford.edu (M.S.); erankin@stanford.edu (E.B.R.); kateh@stanford.edu (K.C.H.); egraves@stanford.edu (E.E.G.); bwloo@stanford.edu (B.W.L.J.); 2Department of Surgery, Stanford University School of Medicine, Stanford Cancer Institute, Stanford, CA 94305, USA; banita@stanford.edu (B.T.); amutahar@stanford.edu (A.Z.I.M.); 3Department of Comparative Medicine, Stanford University School of Medicine, Stanford, CA 94305, USA; kmcasey@stanford.edu

**Keywords:** breast cancer, breast conservation, lumpectomy, radiotherapy, radiotherapy toxicity, syngeneic, orthotopic murine breast cancer, ultra-high dose rate radiotherapy, FLASH radiotherapy

## Abstract

Radiotherapy is a standard treatment for breast cancer, providing effective tumor control and improving survival rates. However, it often causes side effects that can impact patients’ quality of life. In this study, we evaluated ultra-high dose rate radiotherapy, known as FLASH, which has shown potential to reduce normal tissue damage compared to conventional radiotherapy while maintaining its effectiveness. Using a preclinical breast cancer model, we compared the ability of FLASH and conventional radiotherapy to eradicate tumors. Our results demonstrated that both approaches achieved similar tumor control, with the highest success of eradication in smaller tumors treated at higher doses. These findings support ongoing efforts to advance FLASH research in breast cancer.

## 1. Introduction

Radiotherapy (RT) is an effective treatment for invasive breast cancer (BC) and ductal carcinoma in situ (DCIS) [1,2,3]. Unfortunately, there are associated side effects that can include breast dermal and glandular fibrosis, breast shrinkage, shoulder dysfunction, proportionally higher rates of lymphedema when added to any form of lymph node surgery, limited options for reirradiation, higher failure rates with implant-based breast reconstruction, and a limited ability to apply RT to larger areas such as whole liver or lung [4,5,6,7,8,9,10]. Williams et al. noted a 47% rate of psychosocial side effects on a 15-point scale in patient-reported outcomes for individuals receiving RT for breast cancer compared with 23% of patients treated overall with RT for malignancy [5]. Radiation oncologists have ameliorated but not eliminated RT toxicity to normal tissue (NT) [11,12]. RT related toxicity to NT accordingly still poses psychological dilemmas for newly diagnosed patients. Some of the most powerful advocates against RT are often previously treated patients who suffer from complications and convey their personal dissatisfaction to those newly diagnosed whom they encourage to avoid radiotherapy [13,14]. RT use will increase in human breast cancer treatment as it is increasingly used with sentinel node biopsy (SNB) in lieu of the far more morbid standard axillary node dissection (ALND) for patients with nodal involvement. Improved forms of RT for human breast cancer could increase the use of breast conservation, reduce treatment-related complications, improve patient-reported outcomes, and further enhance tumor control and survival.

Currently, conventional dose rate radiotherapy (CONV) is delivered at dose rates less than 0.04 Gy/s [15]. FLASH is a relatively new technique that delivers RT at ultra-high dose rates over 40 Gy/s [16]. Evidence from preclinical models has shown that FLASH causes growth delay comparable to CONV with generally far less initial NT toxicity [16,17,18]. Single fraction FLASH to the skin of the hind limb in mice at doses up to 30 and 40 Gy showed less skin toxicity than CONV [19]. In a first in human safety and feasibility study, a single patient with cutaneous T cell lymphoma was treated successfully with FLASH [20] in one dose rather than the CONV hypofractionated therapy. There is a paucity of preclinical FLASH studies assessing its merits against breast tumors and none utilize in vivo models assessing tumor eradication [21,22,23,24,25,26]. It is critical to further assess the relative benefits and risks of FLASH vs. CONV to determine if FLASH should advance towards use in human breast cancer. As a trial using single fraction FLASH to treat squamous cell carcinoma in cats was associated with late toxicity to the bone, FLASH may also have limitations with toxicity with longer follow-up [27]. These studies further support the need for ongoing investigation into tumor control and normal tissue toxicity.

Multiple studies have shown that breast cancer is often associated with occult multifocal and/or multicentric disease [28,29]. Radiotherapy has proven its worth in eradicating residual, occult disease after lumpectomy in several randomized clinical trials in humans and accordingly is a standard of care for most patients undergoing lumpectomy to reduce in-breast recurrences [30,31,32,33,34,35]. In these early studies, the rates of in-breast recurrence ranged from 7.5% to 14.3% with follow up extending from 5 to 20 years. More recent randomized trials of breast conservation therapy also using whole breast irradiation have demonstrated much lower in-breast recurrence rates such as 0.4% at 5.8 years in one randomized study [36], with another randomized trial showing a recurrence rate of 3.9% at 10 years [2]. Decreased in-breast recurrence rates are likely a result of several factors, including improved preoperative imaging with associated changes in surgery and/or patient selection such as with breast MRI, improved systemic therapy, and advances in radiation treatment planning [37,38]. With respect to breast MRI, considered the most sensitive test for identifying occult breast cancer [37], we previously found that that the most common reason for false positive findings with breast MRI was lesions under 4 mm in size [39]. This suggested to us that 3 mm was a reasonable baseline to use in assessing the effectiveness of radiotherapy for eradication of subclinical, occult breast cancer remaining after lumpectomy in the current era of breast conservation where MRI is commonly used prior to breast conservation surgery/lumpectomy.

Initial experience with FLASH vs. CONV in breast cancer is promising. A preclinical study using a xenograft model with HBCx-12a BC tumors, an aggressive TNBC phenotype [16], demonstrated delay in tumor growth, but not eradication. Growth delay, akin to a partial clinical response after neoadjuvant chemotherapy, is not a clear surrogate for eradication. Eradication is a necessary oncologic endpoint for successful radiotherapy. The most common assessment of comparative tumor eradication between different radiotherapy techniques is tumor control dose 50 (TCD50), which refers to the radiation dose required to achieve local control of a tumor in 50% of the cases. Such studies require many mice. The only study to date evaluating TCD50 was performed in a heterotopic, subcutaneous (hind limb), mammary carcinoma mouse model with tumor volumes of 200 mm^3^ using proton pencil beam, and showed comparable TCD50 of around 50 Gy for both FLASH and CONV [40]. To further advance knowledge of the ability of FLASH to control tumor growth in a more clinically relevant scenario, there is a rationale for testing FLASH using complete eradication of all tumors as an endpoint and using an orthotopic location.

A separate study using a murine model of glioblastoma demonstrated that tumor eradication with FLASH led to long-term immunity to subsequent tumor challenge [41]. This further suggests merit to identifying techniques for orthotopic, single-fraction tumor eradication as might be used after lumpectomy as a potential opportunity for induction of long-term immunity.

For this pilot study, we focused our efforts on eradication of small tumors of ~3 mm in maximal dimension at the time of treatment (20–40 mm^3^) but also treated larger tumors 250–800 mm^3^. The latter tumor sizes are more in keeping with results from other preclinical studies assessing growth delay, but do not replicate the way RT is most used to treat humans. We hypothesized that if FLASH was as effective as CONV in controlling small tumor nodules as is performed routinely in clinical practice in this pilot study, approximating residual occult disease after lumpectomy in human breast cancer, this would provide significant justification for further evaluation of FLASH towards use in human breast cancer. We also hypothesized that a pilot study of FLASH would confirm the effectiveness of CONV in producing growth delays associated with a temporary reduction in tumors size with larger tumors as seen by other investigators in preclinical breast cancer tumor models.

## 2. Materials and Methods

### 2.1. Animals

All animal experiments and procedures were approved by the Institutional Animal Care and Use Committee of Stanford University in accordance with institutional and NIH guidelines. Six- to eight-week-old female C57BL/6 mice were obtained from the Jackson Laboratory (Bar Harbor, ME, USA). Standard animal care and housing were provided by the Stanford University School of Medicine and are under the care and supervision of the Department of Comparative Medicine’s Veterinary Service Center. Mice were maintained on the irradiated Envigo Teklad diet containing 18% protein and 6% fat.

### 2.2. Orthotopic Mouse Models

A radiation sensitive mammary tumor cell line, Py117, derived from the transgenic model of the mouse mammary tumor virus promoter driving the polyoma middle T antigen (MMTV-PyMT), was chosen as the tumor model. Py117 efficiently forms non-metastatic orthotopic tumors in C57BL/6 mice [42]. Py117 cells were cultured using F12K media containing 5% fetal clone II (Hyclone), MITO (1:1000 dilution, BD Biosciences, San Jose, CA, USA), 50 mg/mL gentamicin, and 2.5 mg/mL amphotericin B [42]. Injections were prepared with 10^6^ cells suspended in 50 µL sterile PBS. Ten- to eleven-week-old healthy C57BL/6 female mice (*n* = 67) were inoculated in the left fourth mammary fat pad, which is just beneath the abdominal wall, while under anesthesia (induction 3% and maintenance 1.5% isoflurane in pure O_2_). Tumor inoculations were performed in two sequential experiments. In Round 1, mice were divided into groups with either small tumor sizes (20–40 mm^3^; *n* = 16) or large tumor sizes (250–800 mm^3^; *n* = 12), while in Round 2 only small-size tumors were irradiated (20–40 mm^3^; *n* = 32). For mice in the “small” tumor cohorts, tumor nodules were allowed to grow for a week, while the larger tumors were allowed to grow for 4 weeks. A detailed experimental design for both rounds is outlined in Table 1.

### 2.3. Tumor Volume Measurements

Tumor measurements were acquired without anesthesia. Mice were scruffed and tumor measurements were acquired using calipers. Tumor volumes were derived from two orthogonal measurements using the ellipsoid approximation formula (volume = 0.5 × length × width^2^). Tumor measurements were initiated when tumors became palpable, and thereafter, three days per week. Mice were euthanized at the first sign of ulceration per protocol conditions for this pilot study.

### 2.4. Euthanasia Criteria

If any mouse developed ulceration in the tumor area, it was euthanized. Furthermore, if more than one mouse within the same dose group experienced ≥50% ulceration (with or without alopecia) in the irradiated field, the entire group was terminated without providing skin treatment.

### 2.5. Dose Target Groups

In Round 1, small tumor cohorts were treated with single fractions of 20 or 30 Gy target dose of FLASH or CONV (*n* = 4 per group). The large tumor cohort was treated with a single fraction of 30 Gy target dose of FLASH or CONV (*n* = 6 per group). Seven control mice were left untreated and used to confirm tumor growth in the absence of irradiation. In Round 2, only small-size tumors were treated (*n* = 36). The small-size groups treated in Round 1 with 20 or 30 Gy target dose with either FLASH or CONV were also repeated in Round 2 (*n* = 4 per group). Additionally in Round 2, a target dose group at a dose midpoint between 20 and 30 Gy was treated with 25 Gy target dose of FLASH or CONV (*n* = 8 per group). A detailed experimental design for both rounds is outlined in Table 1.

### 2.6. Stereotactic Mouse Positioner and Beam Collimator

During positioning and irradiation, mice were anesthetized with a mixture of ketamine (100 mg/kg) and xylazine (10 mg/kg) injected into the peritoneum. Immediately after irradiation, mice were placed on a warming blanket until they recovered from anesthesia. They were then returned to their standard housing environment. All 3D prints were designed in Fusion 360^®^ v.2023 (Autodesk, San Rafael, CA, USA). 3D computer-aided design (CAD) files were edited with Ultimaker Cura v.4.3.1 and printed with Ultimaker S5^®^ (New York, NY, USA) using polyactic acid (PLA).

The mice were placed in a 3D-printed stereotactic mouse positioner with a 1.5 × 2.0 cm lateral opening. The mice were immobilized, allowing part of the abdominal wall tissue to protrude outside the wall of the positioner (Figure 1A). The tumors were centered at the lower part of the lateral opening of the mouse stereotactic positioner and the lower bodies of the mice were gently pushed towards the opening of the positioner using paper tissue. The stereotactic mouse positioner is placed on the top of the beam collimator with a 2.0 × 2.0 cm radiation exposure field (Figure 1B). The stereotactic mouse positioner interlocks on the top of the collimator, which provides serial 2.5 mm lateral slots across the field of irradiation. This setting allows for irradiation exposures at different depths within the stereotactic positioner, providing an option for various treatment margins (Figure 1C; Table 1). The beam collimator is designed for electron irradiation geometry with floor-to-ceiling beam orientation, with a 3 cm thick layer of aluminum oxide powder (Al_2_O_3_; 99.99% trace metals basis; Sigma-Aldrich, St Luis, MO, USA) for electron attenuation, in tandem with a 1 cm thick layer of tungsten spheres (2 mm diameter; Tungsten Parts Wyoming, Laramie, WY, USA) for reduction in Bremsstrahlung radiation produced from the collimator’s material (0.2% Bremsstrahlung radiation leakage; Figure 1D).

### 2.7. Irradiation Study Design

Tumor-bearing mice (*n* = 60) were irradiated with either FLASH (94, 193, or 200 Gy/s) or CONV (0.136 or 0.146 Gy/s) dose rates using approximately 16 MeV (E_0_ = 16.6 MeV FLASH and 15.7 MeV CONV; Table 2) electron beams from a previously described configuration of a clinical Varian Trilogy [43,44] and a microcontroller-based (Red Pitaya, Slovenia, Europe) pulse control methodology [43]. Radiotherapy treatments were administered in two sequential distinct experiments, Round 1 and 2, with some key differences between them: (a) in Round 1 the source-to-surface distance (SSD; scattering foil to mouse surface) for FLASH beam geometry was 18.7 cm compared to 76.1 cm SSD for the CONV beam geometry (Figure 1E, CONV vs. FLASH notation), as described previously [45,46]. In Round 2, due to improvements in our configuration to the beam geometry, the mismatch was resolved and all experiments thereafter implemented the same geometry between modalities (Figure 1E, CONV* vs. FLASH notation) [47]. (b) In Round 1, a 7.5 × 20 mm^2^ abdominal wall treatment field (AWTF; Table 1) was used, providing an approximate minimum treatment margin to the tumor of 3.75 mm, which is defined as the minimum distance between the tumor epicenter and the edge of the radiation treatment field (Figure 1C(i)). In Round 2, a 5 × 20 mm^2^ abdominal wall treatment field was employed (Figure 1C(ii); Table 1), yielding an approximate minimum treatment margin to tumor of 2.5 mm. For simplicity, the minimum treatment margin to tumor distance will henceforth be referred to as “treatment margin” throughout the manuscript and figures. (c) In Round 1, only an even number of Gy were used in the target dose groups (20 and 30 Gy) and, therefore, 2 Gy pulses were used, while in Round 2, a dose midpoint 25 Gy dose target group was introduced that required implementation of 1 Gy pulses. Table 1 shows all experimental groups and their associated beam geometry parameters, and Table 2 outlines the beam characteristics and the pulse delivery design for each modality.

### 2.8. Dosimetry

Absolute target doses were determined at the surface of the beam collimator using radiochromic film (EBT3 Galfchromic, Ashland Inc., Wayne, NJ, USA; Figure 2A,B) and exit charge measurements at the Bremsstrahlung tail of the electron beam using an ion chamber (Farmer^®^ Chamber, PTW Model TN30013, Boonton, NJ, USA; Figure 1E). This reference point aims to represent the tumor’s entrance surface dose. Prior to the experiment, films and chamber measurements were used to calibrate target doses of FLASH and CONV. During each mouse irradiation, one 2.4 × 5.1 cm piece of radiochromic film per mouse was placed under the stereotactic mouse positioner, on the top of the beam collimator, centered on the 2.0 × 2.0 cm irradiation field (Figure 2C). Films were allowed to self-develop for more than 24 h post-irradiation before being scanned at 72 dots-per-inch resolution. An average optical density (OD) of the irradiated area was converted to an absorbed dose using a predetermined relationship of OD vs. dose gradient according to the recommendations of the manufacturer [48]. Subsequently, using the associated exit charge reading from the ion chamber for each mouse, an average of nC/Gy was determined for each group. The absolute dose per mouse was then calculated using the exit charge reading, which reduced the inherent variation in the film readings (<0.5% variation in ion chamber readings vs. <3% from radiochromic film). The beam profiles of the collimation were assessed at the surface of the collimator in X (transverse) and Y (craniocaudal) direction for FLASH and CONV, and both CONV geometries (CONV and CONV*; Figure 1E) using experimental films (Figure 2B–E). The percentage dose depth (PDD) distributions of the beam were assessed for both FLASH and CONV geometries using sagittal films, oriented parallel to the beam (Z direction), sandwiched between solid water (Figure 2F).

### 2.9. Beam Homogeneity and Dose Distribution

The examination of the experimental films’ transverse and craniocaudal profiles revealed a symmetrical distribution in the X and Y axes for both FLASH and CONV geometries, as depicted in Figure 2D,E. In the first round of irradiations, the initial set up featuring varying beam geometries—76.1 cm for FLASH and 18.7 cm for CONV surface distance from the scattering foil, as shown in Figure 1E—the full width half maximum (FWHM) of the profiles corresponded closely with the predefined field dimensions: 4.2% and 3.8% for FLASH, and 2.3% and 2.1% for CONV in the X and Y directions, respectively (Figure 1E and Figure 2D). A notable 14.5% difference in dose profiles at the field’s perimeter prompted the incorporation of a wide abdominal wall treatment field of 7.5 × 20 mm^2^ within the mouse stereotactic positioner, yielding a minimal 0.9% dose variance at the positioner’s transverse aperture, where the tumor protrudes (Table 1; Figure 1C(i) and Figure 2D).

During the second round of irradiation, beam configurations were harmonized for both FLASH and CONV to a 19.2 cm surface distance from the scattering foil (Figure 1E). Here, the FWHM differences measured 3.2% and 3.5% for FLASH, and 3.5% and 3.4% for CONV along the X and Y axes, respectively, aligning with the stipulated field size (Figure 1E and Figure 2E). At this stage, the dose discrepancy at the radiation field’s edge was reduced to 4.2%. Consequently, a 5 × 20 mm^2^ abdominal wall treatment field was established, which exhibited a 1.7% difference in dose between the FLASH and CONV modalities at the transverse entry point of the positioner where the tumor emerges (Figure 1C(ii) and Figure 2E).

### 2.10. Radiation Doses

Figure 2A illustrates the film-derived, charge-weighted absorbed doses on the surface of the radiation shield. There was slight dose variance within the groups irradiated separately in Round 1 and 2 (with four mice per round undergoing 20 and 30 Gy target dose treatments on smaller tumors) when considering FLASH or CONV treatments individually. However, the overall dose levels between the FLASH and CONV groups were in alignment. When averaging the measured doses from both rounds for small tumors (this includes the 20 Gy and 30 Gy cohorts that were irradiated in two rounds, as well as the 25 Gy cohort that was irradiated in one round), the absorbed doses were closely matched between FLASH (19.9 ± 0.24, 24.0 ± 0.11, and 30.0 ± 0.73 Gy) and CONV (20.2 ± 0.51, 24.6 ± 0.08, and 30.0 ± 0.58 Gy) modalities, with a difference in less than 3% noted. A more pronounced variation of 4.5% was observed in the large tumor-bearing mice targeted with 30 Gy dose, where the FLASH group had an average absorbed dose of 28.6 ± 0.25 Gy in comparison to the CONV group’s 30.0 ± 0.06 Gy. In the target dose groups of 20 and 25 Gy, one specimen from each group received one fewer 2 Gy pulse than prescribed, as can be discerned in Figure 2A (indicated by red arrows). Mice receiving a lower number of pulses were omitted from subsequent analysis reported below.

### 2.11. Statistics

Tumor volumes were compared between FLASH- and CONV-irradiated groups at each post-irradiation time point using the Mann–Whitney U test, with multiple comparisons corrected using the Holm–Šidák method and statistical significance defined as α = 0.05. Due to the non-parametric distribution of tumor volume data and the small sample size, confidence intervals around median differences were not directly computed. Statistical analyses and graphical representation of data were performed using GraphPad Prism (Version 10.0, GraphPad Software, San Diego, CA, USA). The 20 Gy and 30 Gy groups were irradiated separately in Round 1 and Round 2 (*n* = 4 per group per round), with slight variations in delivered dose and treatment margins between rounds. To address these variations, analyses were initially stratified by irradiation rounds to verify the consistency of results before data from both rounds were merged to evaluate overall treatment effects. The 25 Gy groups, irradiated only in Round 2, were analyzed independently. Detailed results of Mann–Whitney U tests, including exact *p*-values and mean rank differences for each time point comparing FLASH and CONV groups, are provided in Appendix A.

## 3. Results

### 3.1. FLASH Shows Comparable Tumor Control to CONV Against Small Tumors at 20, 25, and 30 Gy, with 30 Gy Eradicating Most Small Tumors

Smaller tumors, averaging 30 mm^3^ and treated with a single 20 or 25 Gy dose, exhibited similar levels of tumor regression with FLASH and CONV with tumor eradication in some and tumor regrowth in others. Mice were euthanized at the initial sign of ulceration without treatment. Autopsies were not performed on mice found dead. From the total of 30 tumors (*n* = 7 per group for 20 Gy and 25 Gy FLASH, *n* = 8 per group for 20 Gy and 25 Gy CONV), 26 tumors initially responded with complete remission (unmeasurable) while one tumor per group remained palpable. Overall, tumors that were not eradicated regrew to <8 mm^3^ within 21 to 28 days of irradiation (Figure 3A,B). For the 20 Gy irradiation groups, 29% of tumors receiving FLASH and 43% tumors receiving CONV (one death, etiology unknown) remained in complete remission on day 46. The remainder grew past their original volume, with one exception from each group that had half the original volume (Figure 3A). At the same time point, for the 25 Gy FLASH group, 83% of tumors (one death, etiology unknown) regrew to their original volume, with the exception of one tumor that had 60% of its original volume, while 50% (two deaths, unknown etiology) for the 25 Gy CONV group regrew above their original volume, with the exception of two tumors that had < 50% of their original volume (Figure 3B). No statistically significant differences were observed between FLASH- and CONV-irradiated tumors at any time point after irradiation for both the 20 Gy and 25 Gy groups (Appendix A). Specifically, for the 20 Gy group, the mean rank difference in tumor volumes between FLASH and CONV at day 46 was 0.464 mm^3^ (Mann–Whitney U test, *p* = 0.848), indicating no significant difference. Similarly, for the 25 Gy group, the mean rank difference was 3.333 mm^3^ (*p* = 0.132), also showing no statistical significance. Notably, tumors treated with 20 Gy had a wider treatment margin in Round 1 (3.75 mm) compared to Round 2 (2.5 mm), whereas tumors treated with 25 Gy were all treated in Round 2, consistently receiving a 2.5 mm treatment margin. The 20 Gy data are presented as a pooled dataset from both rounds; however, independent analyses were also conducted separately for Round 1 and Round 2, showing no statistically significant differences between FLASH and CONV at any time point post-irradiation (Appendix A). There was no clear cause of death in the mice that died during the study period as they exhibited neither skin toxicity nor signs of gastrointestinal toxicity such as diarrhea.

Treatments with a 30 Gy single dose, like the 20 Gy group, were also carried out in two rounds, as presented in Figure 3C,D. In Round 1, the disparate beam geometries for FLASH and CONV required a 7.5 × 20 mm^2^ abdominal wall treatment field (3.5 margin) to account for the difference at the field edge (Table 1). Both modality groups (*n* = 4 per group) exhibited complete remission by two weeks, which persisted up to four weeks. Mice were euthanized at the first sign of ulceration per protocol conditions for this pilot study (Figure 3C). In Round 2 with matched beam geometry, in order to minimize radiotoxicity, a smaller 5 × 20 mm^2^ abdominal wall treatment field (2.5 mm margin) was pursued (Table 1). Following the 30 Gy treatment with a 2.5 mm margin, both modality groups (*n* = 4 per group) exhibited complete remission by two weeks. By day 48, 50% of tumors from the FLASH group and 66% of tumors (one death, etiology unknown) from the CONV group remained in complete remission. At the same time point, one tumor from the FLASH group had regrown past its original volume, and one had regrown to 75% of its original volume, while in the CONV group, one mouse’s tumor had regrown to its original size (Figure 3D). Histological verification indicated that one nodule in the FLASH group was not a tumor as suspected (Appendix A) but rather an enlarged lymph node, as shown in Appendix A. Regardless of the histological adjustments, there was no significant difference in tumor growth between the 30 Gy FLASH and CONV groups. No statistically significant differences were observed between 30 Gy FLASH- and 30 Gy CONV-irradiated tumors at any time point after irradiation for both 3.75 mm treatment margins at Round 1 and 2.5 mm for Round 2 (Appendix A). Specifically, for the 30 Gy group at Round 1 (Figure 1C), the mean rank difference in tumor volumes between FLASH and CONV at day 33 was 0 mm^3^ (Mann–Whitney U test, *p* > 0.999), indicating no significant difference. Similarly, for the 30 Gy group on Round 2 (Figure 3D) at day 46, the mean rank difference was 0.583 mm^3^ (*p* = 0.829), also showing no statistical significance.

Among the 20, 25, and 30 Gy doses, 30 Gy achieved the highest tumor eradication rate but also produced the most pronounced skin toxicity. Specifically, administering 30 Gy with a 3.75 mm treatment margin caused greater abdominal wall skin exposure—thus limiting follow-up beyond 30 days—compared to using a 2.5 mm margin.

### 3.2. FLASH Is as Effective as CONV in Delaying Growth of Larger Tumors

In the initial week, unirradiated Py117 orthotopic tumors typically reach a volume of about 30 mm^3^, escalating to ~635 mm^3^ by the fourth week, and surpassing 1000 mm^3^ in the fifth week, as shown in Figure 4A. While smaller tumors display a consistent size range, larger ones exhibit greater variability in size. Larger tumors, with mean volumes of 493.05 ± 209.04 mm^3^ in the FLASH group and 497.71 ± 195.72 mm^3^ in the CONV group, exhibited similar regression within the first two weeks following treatment with a 30 Gy dose with either FLASH or CONV radiation, followed by a parallel pattern of regrowth. No statistically significant differences were observed between the two modalities at any time point post-irradiation (Appendix A). Specifically, by day 16, the mean rank difference in tumor volumes between FLASH and CONV was −0.667 mm^3^ (Mann–Whitney U test, *p* = 0.818), and by day 23, it was −1.333 mm^3^ (Mann–Whitney U test, *p* = 0.589), indicating no significant difference. Despite the FLASH group receiving a slightly reduced dose of 28.6 Gy compared to the 30 Gy for the CONV group, tumor control was equivalent between the two modalities. The study concluded after three weeks with initial signs of skin ulceration or other skin toxicity associated with large tumors toxicity, such as alopecia, affecting three mice in the FLASH group and four in the CONV group. It was determined that a 30 Gy irradiation dose was insufficient for the complete eradication of larger tumors, as depicted in Figure 4B.

## 4. Discussion

Radiotherapy is an important treatment modality for many patients with breast cancer [20]. The role of radiotherapy in breast cancer care is so broad that there are projected to be 2.01 million breast cancer survivors treated by radiotherapy by 2030 [49]. Further efforts to reduce toxicity associated with radiotherapy while maintaining effectiveness would represent a significant medical advance [50,51]. Early clinical studies have been initiated. FAST-01, a feasibility study applying FLASH to bone metastases in ten patients with short term follow-up has been completed [52]. Other studies in humans are underway, such as FAST-02, a trial aimed at exploring thoracic bone metastases [53], a Phase I trial determining the maximum tolerated dose of FLASH for melanoma skin metastases [54]; and, a randomized Phase II trial comparing FLASH and conventional radiotherapy for treating basal and squamous cell skin cancers [55,56]. At the time this manuscript was written, there were no human trials identified in clinicaltrials.gov applying FLASH to primary human breast cancer.

While many studies have demonstrated less normal tissue toxicity and growth delay when irradiating breast tumors with FLASH vs. CONV, we wanted to determine whether FLASH could also achieve the more meaningful clinical endpoint of tumor eradication. Based on prior data from our group, which demonstrated that FLASH showed less normal tissue toxicity than CONV, that the maximum tolerated single fraction dose of CONV was in the range of 20 to 30 Gy, and that tumor response is dose-related, we compared FLASH vs. CONV at 20, 25, and 30 Gy with small tumor nodules, as might remain after lumpectomy, vs. larger tumors as have been assessed in other preclinical studies. We hypothesized that radiotherapy would have the potential to eradicate small tumor nodules, as is known to occur clinically when post-lumpectomy radiotherapy is used to eradicate occult residual disease, while also hypothesizing that radiotherapy of larger tumors would likely show growth delay, as has been shown in other preclinical studies. In order to assess the response of both small and large tumors in a common, orthotopic location, we elected to grow tumors in the left fourth mammary fat pad as this was the only orthotopic location that could accommodate both small and large tumors without impairing the mobility of study mice.

We deliberately chose the Py117 tumor cell line in C57BL/6 mice as this model has been used in our labs and has been found to have a low metastatic potential, thus, allowing focused investigation of local tumor control without the competing risk of losing study mice to metastatic disease. We did not seek to assess normal tissue toxicity, as the location of the fourth mammary fat pad is not clinically relevant for breast cancer, nor did we seek to assess for metastatic disease, as this is more related to the biology of the primary tumor and dissemination of the tumor prior to radiotherapy.

This dose–response study ultimately showed equivalent tumor control with single-fraction doses of FLASH compared to CONV across all doses and tumor sizes. We also demonstrated comparable eradication rates for both FLASH and CONV of small breast cancer nodules up to 48 days, with 30 Gy demonstrating the highest eradication rate. Both FLASH and CONV produced similar growth delay in larger breast tumors, consistent with the results of others. Eradication of small breast cancers, therefore, is possible, and is perhaps a more meaningful endpoint to consider for future studies of FLASH than growth delay alone.

Interestingly, while we noted the most pronounce skin toxicity with 30 Gy in this study, our group previously did not see lethal skin toxicity at 30 Gy when radiotherapy was applied to the hind limb [19]. We also failed to see skin toxicity in a model of unilateral, left-chest wall irradiation with 6 months of follow-up (unpublished). Additionally, in our study, mice were euthanized at the initial sign of ulceration without treatment, and this limited the extent of the follow-up. Therefore, considering the ideal placement of orthotopic tumors for investigating tumor eradication, the third mammary fat pad, together with appropriate skin treatments for acute radiation toxicity, would facilitate reliable long-term monitoring by mitigating the limitations posed by skin toxicity.

The role of treatment margins underlines the importance of beam geometry in dose delivery. The initial larger margin used with different beam geometries for FLASH and CONV highlighted the necessity of precise dosimetry. This is particularly important at the field edge where dose fall-off is critical for sparing normal tissue while ensuring adequate tumor coverage. The design of the experimental geometry in our study was necessary to address the potential for GI toxicity. By orienting the FLASH beam from floor to ceiling and positioning the mice laterally, we strategically targeted the tumor-bearing fourth mammary fat pad. This approach ensured that the tumors were within the radiation field while minimizing the abdominal exposure to radiation. While reducing the tumor irradiation margin could reduce the skin toxicity and gastrointestinal exposure, it increased the possibility that tumors might not be irradiated in their entirety, which likely contributed to more recurrences in the cohort of small tumors receiving 30 Gy treated with the 2.5 mm margin. This echoes the delicate balance that must be struck in radiation therapy between tumor eradication and the preservation of quality of life through the mitigation of treatment-related side effects.

Autopsies were not performed on mice found dead and this limited the knowledge of cause of death. Additionally, in working with smaller tumors, we found that assessment of nodules below 50 mm^3^ was potentially misleading due to misinterpretation of enlarged lymph nodes as tumor recurrence in a few study animals, confirmed in pathology. The absence of statistical significance for anti-tumor effectiveness within small and large tumor cohorts between FLASH and CONV remained even after accounting for histological corrections, and the exclusion of palpable lymph nodes from tumor volume measurements. Tumor growth delay between FLASH and CONV for large-size tumors was comparable, validating previous findings [16].

Quantitative skin toxicity experiments were not performed in this study because we have previously demonstrated skin radiotoxicity benefits of FLASH versus CONV radiation in the area of the thoracic cavity [19]. Considering that in the current study we have irradiated tumors adjacent to the abdominal cavity, we have encountered losses in mice possibly due to the radiotoxicity of the GI track, which cannot be confirmed due to the lack of postmortem biopsies. Currently, the FLASH configurations and geometries are limited to a single beam angle, sustaining relevant dose rates. Ideas of different configurations like FLASH EXACT [57,58] that would introduce conformal FLASH radiotherapy would allow higher doses to the tumor without the limitation of the toxicity and even higher doses could be tested on mice without concerns of radiotoxicity. Stereotactic regimens using multiple tangents may also produce tumor control with less toxicity [31]. A recent pilot study comparing single fraction vs. hypofractionated radiotherapy did not, however, demonstrate improved tumor control or less toxicity, though a dose of only 20 Gy was used and tumor growth delay was the endpoint rather than tumor eradication [41].

The radiobiology of FLASH vs. CONV in breast cancer, as well as other solid tumors, remains incompletely understood [18,59,60] despite ongoing efforts in this direction [41,45,61].

## 5. Conclusions

Investigator-initiated studies have shown that occult residual disease commonly remains after lumpectomy for breast cancer in humans even with adequate tissue margins. Randomized trials have demonstrated that radiotherapy is able to successfully eradicate most of these residual, occult foci. Despite its benefits, radiotherapy is associated with short- and long-term toxicities that limit the number of individuals who choose breast conservation and/or significantly complicate quality of life for many. FLASH represents an opportunity to improve the quality of life for millions of women and men if proven to be safer, yet as effective, as current dose-rate radiotherapy.

In keeping with the findings of other investigators, we confirmed FLASH is able to produce comparable growth delay in larger tumors. However, we also demonstrated for the first time, to our knowledge, that single dose FLASH has comparable eradication rates for small breast cancer foci equivalent to conventional dose-rate radiotherapy in a syngeneic, orthotopic murine model of breast cancer. Future studies will improve our understanding of ultra-high dose rate radiotherapy effectiveness and toxicity as a possible new paradigm in treating human breast cancer.

## Figures and Tables

**Figure 1 cancers-17-01095-f001:**
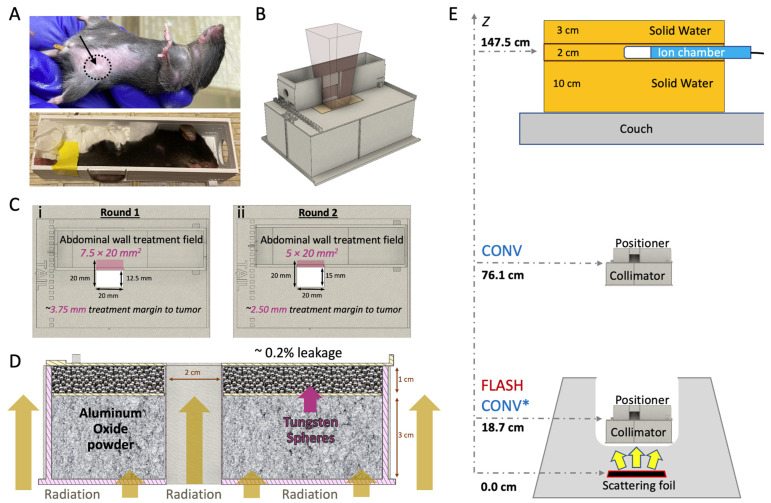
In vivo breast cancer orthotopic tumor irradiation with FLASH or CONV dose rates. (**A**) Breast cancer orthotopic tumor-bearing mouse with approximately 30 mm^3^ tumor at the fourth mammary fat pad (top; black dashed circle and arrow). An anesthetized mouse placed inside the mouse positioner frame with the tumor centered at the bottom of the lateral opening and immobilized with paper tissue (bottom). (**B**) 3D computer-aided design (CAD) files illustrating the positioning of the mouse positioner on the collimator, the positioning of the radiochromic film during irradiation. (**C**) Top view of the collimator featuring the mouse positioner in two treatment margins to tumor: (i) Round 1, positioned to expose a 7.5 × 20.0 mm^2^ area of the abdominal wall tissue, 3.75 mm margin or (ii) Round 2, positioned to expose a 5.0 × 20.0 mm^2^ area of the abdominal wall tissue, 2.5 mm margin. (**D**) 3D CAD file of the 2.0 × 2.0 cm^2^ collimator presented with a lateral cross-section, illustrating mouse radiation shielding. The collimator is filled with a 3 cm layer of aluminum oxide to stop the electrons and a 1 cm layer of tungsten spheres (2.0 mm diameter) to absorb Bremsstrahlung radiation and efficiently shield the rest of the animal’s body (Bremsstrahlung radiation leakage ~0.2%). (**E**) FLASH and CONV beam geometries. For FLASH irradiations, the collimator and mouse positioner frame are placed inside the treatment head and the beam entrance surface of the mouse is 18.7 cm from the scattering foil. For CONV irradiations, at Round 1, the beam entrance surface of the mouse was 76.1 cm (unmatched geometries) from the scattering foil, and at Round 2 at 18.7 cm (CONV* notation matching geometry with FLASH). The pulses delivered are monitored using an ion chamber, measuring the Bremsstrahlung tail at 11.0 cm solid water depth, and are located 147.5 cm from the scattering foil.

**Figure 2 cancers-17-01095-f002:**
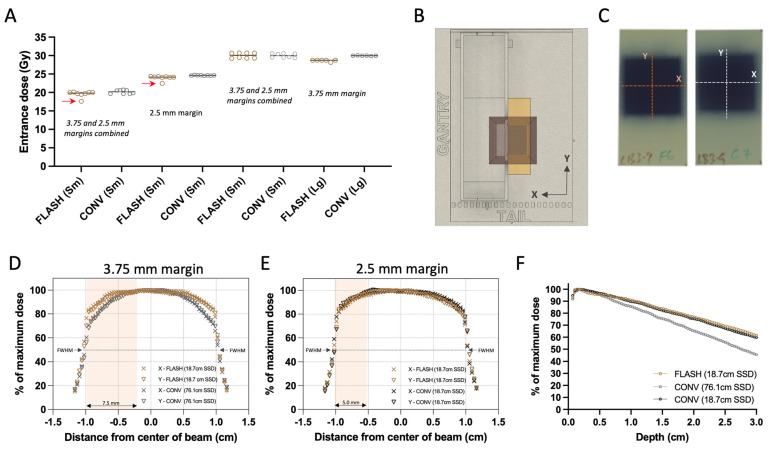
Radiochromic film dosimetry of delivered doses and film-derived collimator characterization. (**A**) Grouped scatter plot of film-derived mean dose of all animals combined (Sm for small 20–40 mm^3^ and Lg for large 250–800 mm^3^ tumor volumes). Red arrows indicate missed pulses, which resulted in elimination of these animals from the analysis. (**B**) Top view of experimental set up illustrating the film positioning and exposure. (**C**) Representative exposed films with either FLASH or CONV 25 Gy single fraction target dose group, illustrating the direction of the beam profiles. (**D**) Film-derived X (transverse) and Y (craniocaudal) profiles of the unmatched geometries between FLASH and CONV with 7.5 × 20 mm^2^ abdominal wall treatment field used in Round 1 of irradiation, and (**E**) profiles from the matched geometries with the 5 × 20 mm^2^ abdominal wall treatment field used in Round 2. Salmon highlights represent the exposure of the abdominal wall in the X direction. (**F**) Film-derived percentage depth-dose curve (PDD) from FLASH and both geometries of CONV configurations, using films parallel to the direction of the irradiation beam. Overall, the delivered doses between FLASH and CONV were comparable and despite the difference in source-to-surface (SSD), profiles and PDDs of the two modalities remain comparable within the tumor volumes.

**Figure 3 cancers-17-01095-f003:**
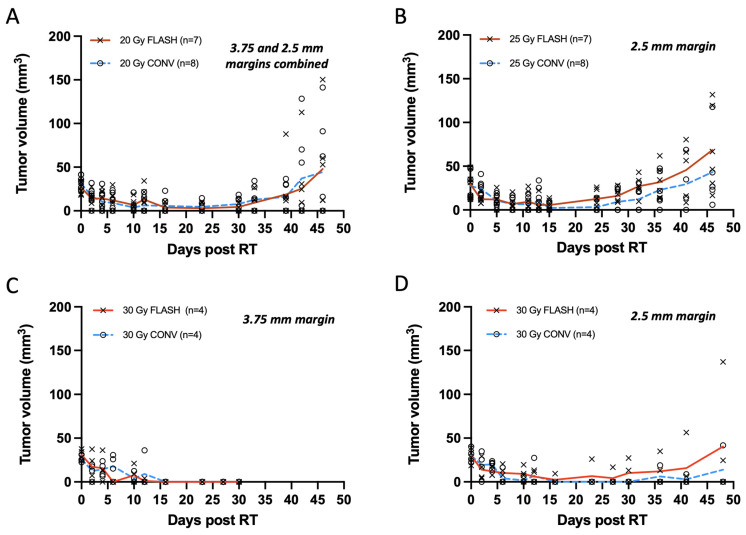
(**A**–**D**) Tumor measurements with calipers plotted as response curves of breast cancer orthotopic tumors of small volume (20–40 mm^3^) irradiated with 20, 25, and 30 Gy single fraction with either FLASH or CONV dose rates. (**A**) Tumor volumes from animals irradiated with 3.75 and 2.5 mm treatment margin combined (*n* = 7 FLASH, *n* = 8 CONV). FLASH group had one animal excluded due to a missed pulse. Tumors were controlled for the first 4 weeks and regressed thereafter, with no significant differences between groups at any time point post-irradiation (Mann–Whitney U test, all *p* > 0.05). (**B**) Tumors targeted with 25 Gy and 2.5 mm treatment margin (*n* = 7 FLASH, *n* = 8 CONV); one FLASH exclusion. Tumors remain controlled for 4 weeks and regressed thereafter, with no differences between groups (Mann–Whitney U test, all *p* > 0.05). (**C**) Tumor treated with 30 Gy and 3.75 mm treatment margin (*n* = 4 per group) were controlled by day 30, but severe tissue toxicity led to study termination. No significant differences observed (all *p* > 0.05). (**D**) Tumors treated with 30 Gy and 2.5 mm treatment margin remained controlled for the first 4 weeks (all *p* > 0.05). By day 48, only one tumor per group showed regrowth (mean rank difference = 0.583 mm^3^; Mann–Whitney U test, *p* = 0.829). Overall, there was no significant difference between FLASH and CONV in tumor growth delay or eradication of small tumor volumes (20–40 mm^3^) with single fractions of 20, 25, and 30 Gy.

**Figure 4 cancers-17-01095-f004:**
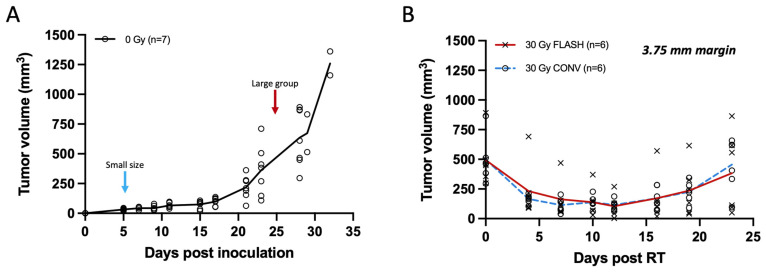
Tumor measurements of breast cancer orthotopic tumors with calipers plotted from non-irradiated controls and tumors of large volume (250–800 mm^3^) irradiated with 30 Gy single fraction with either FLASH or CONV dose rates. (**A**) Tumor growth curve of unirradiated breast cancer orthotopic tumors at the fourth mammary fat pad (*n* = 7). Small size tumor volumes were selected for irradiation on day 5 (20–40 mm^3^; light blue arrow), and larger range of tumor volumes at day 24 (250–800 mm^3^; red arrow). Tumors grow moderately for the first 2 weeks and exponentially thereafter. (**B**) Tumor response curve of breast cancer orthotopic tumors of large volume range irradiated with 30 Gy single fraction of either FLASH or CONV dose rates (*n* = 6 per group). Tumor volumes were suppressed for the first 2 weeks and regressed thereafter with no statistically significant differences observed between FLASH and CONV at any time point post-irradiation (Mann–Whitney U test, all *p* > 0.05). There was no significant difference between FLASH and CONV in tumor growth delay of large tumor volumes (250–800 mm^3^) with a single fraction of 30 Gy.

**Table 1 cancers-17-01095-t001:** Experimental irradiated groups of breast cancer orthotopic tumor-bearing mice, irradiated with either FLASH or CONV dose rates in two irradiation rounds with different abdominal wall treatment fields.

Round 1							
Mode	Tumor Volume	Dose	No. mice	Collimation	AWTF ^a^	SSD ^b^	Missed
	[mm^3^]	[Gy]		[cm^2^]	[cm^2^]	[cm]	Pulses *
FLASH	20–40	20	4	2 × 2	0.75 × 2.00	18.7	1
FLASH	20–40	30	4	2 × 2	0.75 × 2.00	18.7	0
FLASH	250–800	30	6	2 × 2	0.75 × 2.00	18.7	0
CONV	20–40	20	4	2 × 2	0.75 × 2.00	76.1	-
CONV	20–40	30	4	2 × 2	0.75 × 2.00	76.1	-
CONV	250–800	30	6	2 × 2	0.75 × 2.00	76.1	-
**Round 2**							
Mode	Tumor Volume	Dose	No. mice	Collimation	PTV ^a^	SSD ^b^	Missed
	[mm^3^]	[Gy]		[cm^2^]	[cm^2^]	[cm]	Pulses *
FLASH	20–40	20	4	2 × 2	0.50 × 2.00	18.7	0
FLASH	20–40	25	8	2 × 2	0.50 × 2.00	18.7	1
FLASH	20–40	30	4	2 × 2	0.50 × 2.00	18.7	0
CONV	20–40	20	4	2 × 2	0.50 × 2.00	18.7	-
CONV	20–40	25	8	2 × 2	0.50 × 2.00	18.7	-
CONV	20–40	30	4	2 × 2	0.50 × 2.00	18.7	-

^a^ Abdominal wall treatment field; treatment area within the mouse positioner frame (abdominal wall exposure). ^b^ Source to surface distance. * Animals irradiated with missed pulses were excluded from the analysis.

**Table 2 cancers-17-01095-t002:** Beam parameters for experimental beam delivery with FLASH or CONV.

Round 1								
Mode	Rx ^a^	Pulses	DPP ^b^	Pulse Rate	Dose Rate	Pulse Length	IPDR ^c^	Energy
	[Gy]		[Gy]	[Hz]	[Gy/s]	[s]	[Gy/s]	[MeV]
FLASH	20	10	2	90	200	3.75 × 10^−6^	5.33 × 10^5^	16.60
FLASH	30	15	2	90	193	3.75 × 10^−6^	5.33 × 10^5^	16.60
CONV	20	9860	2.03 × 10^−3^	72	0.146	3.75 × 10^−6^	541	15.73
CONV	30	14,796	2.03 × 10^−3^	72	0.146	3.75 × 10^−6^	541	15.73
**Round 2**								
Mode	Rx ^a^	Pulses	DPP ^b^	Pulse Rate	Dose Rate	Pulse Length	IPDR ^c^	Energy
	[Gy]		[Gy]	[Hz]	[Gy/s]	[s]	[Gy/s]	[MeV]
FLASH	20	10	2	90	200	3.75 × 10^−6^	5.33 × 10^5^	16.60
FLASH	25	25	1	90	94	3.75 × 10^−6^	2.67 × 10^5^	16.60
FLASH	30	15	2	90	193	3.75 × 10^−6^	5.33 × 10^5^	16.60
CONV	20	13,200	1.52 × 10^−3^	90	0.136	3.75 × 10^−6^	404	15.73
CONV	25	16,500	1.52 × 10^−3^	90	0.136	3.75 × 10^−6^	404	15.73
CONV	30	19,800	1.52 × 10^−3^	90	0.136	3.75 × 10^−6^	404	15.73

^a^ Rx: prescribed dose (target dose). ^b^ DPP: dose per pulse. ^c^ IPDR: intra-pulse dose rate.

## Data Availability

The datasets used and/or analyzed during the current study are available from the corresponding author on reasonable request.

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
