# Peer review of "Effectiveness of FLASH vs. Conventional Dose Rate Radiotherapy in a Model of Orthotopic, Murine Breast Cancer"

_cancers, 2025, doi:10.3390/cancers17071095_

Round 1

Reviewer 1 Report

Comments and Suggestions for Authors

In this experimental in vivo study using mouse breast cancer model , more specifically a Py117 breast cancer orthotopic tumor and two different electron beam radiation therapy modalities standard (CONV) and high dose rate (FLASH) . The authors using well though and planned experiments show a similar efficiency between CONV and FLASH in eradicating small volume tumors which is something in general very positive. There are though some specific deficits that render this study promising but limited.

The authors do not provide clear picture if this eradication with FLASH is even slightly better. In addition what is the role of dose they have chosen and why?

The lack of survival curves is a huge problem for this and any study. So according to the authors how can we draw specific conclusions on electron beam FLASH and tumor control efficiency which when it comes to patients toxicity is the major limiting factor of therapy efficiency.

No clear picture also on the toxicity and not only skin but long term effects

No clear picture on metastasis in months.

Although this is a valuable study according to this reviewer is highly limited and does not offer any specific and clear insights on the advantage or not of FLASH. One actually can argue strongly on the essence of this results based also on similar studies in Europe like https://pubmed.ncbi.nlm.nih.gov/39293721/ and other not to mention protons of course.

In general and based on current evidence especially in vivo it seems that FLASH does limit toxicity while tumor control maybe similar compared to CONV. 

Author Response

We sincerely appreciate the reviewer for taking the time to evaluate our work. Your thoughtful insights have not only clarified key aspects of our study but have also raised crucial questions, ultimately enhancing the overall clarity of our data presentation.

Reviewer 1

In this experimental in vivo study using mouse breast cancer model , more specifically a Py117 breast cancer orthotopic tumor and two different electron beam radiation therapy modalities standard (CONV) and high dose rate (FLASH) . The authors using well though and planned experiments show a similar efficiency between CONV and FLASH in eradicating small volume tumors which is something in general very positive. There are though some specific deficits that render this study promising but limited.

  1. The authors do not provide clear picture if this eradication with FLASH is even slightly better. In addition what is the role of dose they have chosen and why?

Response 1. We have not sought to show FLASH is more effective than conventional radiotherapy, only that it has comparable effectiveness, given that most studies have shown significant benefit to normal tissue. The selected doses of 20, 25, and 30 Gy were based on prior results from our group (Soto et al. Ref 19) showing that FLASH RT up to 30 Gy is well tolerated compared with conventional RT. Accordingly, since higher dose is typically associated  with better tumor control, we sought a range of doses that would push boundaries on both tumor control and toxicity. Among the three doses tested in our model; the 30 Gy dose demonstrated the most promising results for tumor eradication providing valuable guidance for future studies. Ultimately, if FLASH is “only” as good as conventional radiotherapy, while better preserving normal tissue and creating less morbidity, this will still represent an opportunity to significantly improve clinical care of humans

  1. The lack of survival curves is a huge problem for this and any study. So according to the authors how can we draw specific conclusions on electron beam FLASH and tumor control efficiency which when it comes to patients toxicity is the major limiting factor of therapy efficiency.
  2. No clear picture also on the toxicity and not only skin but long-term effects
  3. No clear picture on metastasis in months.

We thank the reviewer for highlighting these important points.

Response 2. We acknowledge that survival curves are critical in assessing benefit vs toxicity prior to recommending a new treatment. In this study, we wished to focus primarily on local tumor response of both small as well as larger, orthotopic tumors, a local-regional phenomenon, which is dependent on radiotherapy effectiveness, rather than survival, a systemic phenomenon, more related to tumor biology and possibly toxicity.

As described in the Discussion (original manuscript lines 458-466):

Interestingly, while we noted skin toxicity with 30 Gy FLASH in this study, our group previously did not see lethal skin toxicity at 30 Gy when radiotherapy was applied to the hind limb [19]. We also failed to see skin toxicity in a model of unilateral, left chest wall irradiation with 6 months of follow-up (unpublished). For this study, we elected to use the 4th mammary fat pad in order to assess the anti-tumor effect of FLASH vs CONV with larger tumors in an orthotopic location: it would not have been feasible to grow larger tumors if we had used the 3rd mammary fat pad as this would have impaired the motility of mice and led to euthanasia of study animals before our tests could be completed.

Therefore, the use of the 4th mammary fat pad allowed assessment of tumor eradication, which is novel, as well as  confirming results from assessment of tumor growth delay, as reported in other studies.

Response 3. We agree completely with the reviewer’s concern over long-term toxicity with FLASH. That said, this study was designed specifically as a short-term study to assess tumor control in small and large orthotopic tumors, and as a result necessitated using the 4th mammary fat pad, a higher risk location but the only feasible one for our intended purposes. While not detracting from our novel findings of breast tumor eradication, we have made no assertions accordingly regarding the safety of FLASH RT vs CONV RT with respect to breast tumors.

Response 4. We appreciate the concern over metastases. That said, with our focus on local tumor control, we deliberately chose the Py117 tumor cell line and C57BL/6 strain as this combination has been used by our team previously because of its propensity not to form early metastases (Aguilera, et al https://pubmed.ncbi.nlm.nih.gov/32098769/) and therefore would allow us to assess local tumor control without concern for loss of study animals due to metastases. In addition, metastases, as in humans, is a function of occult spread from the primary orthotopic tumor prior to and unrelated to irradiation of the primary tumor, ie a separate biologic event, and in the absence of systemic agents to suppress metastatic disease it is unclear how the formation of metastases would have impacted our findings on local tumor control though perhaps could have accounted for unexplained loss of study mice.

We have acknowledged this reviewer’s concerns and explicitly addressed these limitations in the revised manuscript's discussion section to ensure readers clearly understand the intent, scope, and interpretation of our current findings.

Line 504-544: While many studies have demonstrated less normal tissue toxicity and growth delay when irradiating breast tumors with FLASH vs CONV radiotherapy, we wanted to determine whether FLASH could also achieve the more meaningful clinical endpoint of tumor eradication. Based on prior data from our group which demonstrated that FLASH showed less normal tissue toxicity than CONV, that the maximum tolerated single fraction dose of CONV was in the range of 20 to 30 Gy, and that tumor response is dose related, we compared FLASH vs CONV at 20, 25, and 30 Gy with small tumor nodules, as might remain after lumpectomy, vs larger tumors as have been assessed in other preclinical studies. We hypothesized that radiotherapy would have the potential to eradicate small tumor nodules, as is known to occur clinically when post-lumpectomy radiotherapy is used to eradicate occult residual disease, while also hypothesizing that radiotherapy of larger tumors would likely show growth delay, as has been shown in other preclinical studies. In order to assess response of both small and large tumors in a common, orthotopic location, we elected to grow tumors in the left 4th mammary fat pad as this was the only orthotopic location that could accommodate both small and large tumors without impairing mobility of study mice.

We deliberately chose the Py117 tumor cell line in C57BL/6 mice as this model has been used in our labs and found to have a low metastatic potential, thus allowing focused investigation of local tumor control without the competing risk of losing study mice to metastatic disease. We did not seek to assess normal tissue toxicity, as the location of the 4th mammary fat pad is not clinically relevant for breast cancer, nor did we seek to assess for metastatic disease, as this is more related to the biology of the primary tumor and dissemination of tumor prior to radiotherapy. 

This dose-response study ultimately showed equivalent tumor control with single-fraction doses of FLASH compared to CONV across all doses and tumor sizes. We also demonstrated for both FLASH and CONV comparable eradication rates of small breast cancer nodules up to 48 days, with 30 Gy demonstrating the highest eradication rate. Both FLASH and CONV produced similar growth delay in larger breast tumors, consistent with the results of others. Eradication of small breast cancers therefore is possible, and perhaps more meaningful, endpoint to consider for future studies of FLASH radiotherapy than growth delay alone.

Interestingly, while we noted the most pronounce skin toxicity with 30 Gy in this study, our group previously did not see lethal skin toxicity at 30 Gy when radiotherapy was applied to the hind limb [19]. We also failed to see skin toxicity in a model of unilateral, left chest wall irradiation with 6 months of follow-up (unpublished). Additionally, in our study mice were euthanized at initial signs of ulceration without treatment and this limited the extent of the follow-up. Therefore, considering the ideal placement of orthotopic tumors for investigating tumor eradication, the 3rd mammary fat pad, together with appropriate skin treatments for acute radiation toxicity, would facilitate reliable long-term monitoring by mitigating the limitations posed by skin toxicity.

  1. Although this is a valuable study according to this reviewer is highly limited and does not offer any specific and clear insights on the advantage or not of FLASH. One actually can argue strongly on the essence of this results based also on similar studies in Europe like https://pubmed.ncbi.nlm.nih.gov/39293721/ and other not to mention protons of course.
  2. In general and based on current evidence especially in vivo it seems that FLASH does limit toxicity while tumor control maybe similar compared to CONV.

Response 5. We appreciate the reviewer’s comments and would like to clarify the primary goal of our study, which was to investigate tumor control—specifically, the potential for tumor eradication in breast cancer using FLASH radiotherapy. While numerous studies (including the one referenced at https://pubmed.ncbi.nlm.nih.gov/39293721/) have demonstrated FLASH’s advantage in preserving healthy tissue, our focus was on whether tumor control, and ultimately tumor eradication, could be achieved at comparable levels. We believe this represents a significant and novel insight that could help establish a new standard for comparing FLASH RT with CONV RT in breast cancer, and we propose tumor eradication as a key endpoint for future research.

Response 6. Furthermore, we concur with the reviewer’s perspective, which aligns with our hypothesis. By demonstrating comparable tumor control in orthotopic breast cancer models, in conjunction with well-documented normal tissue–sparing effects, we suggest that FLASH RT may be a promising strategy for eradicating residual tumors following lumpectomy.

Reviewer 2 Report

Comments and Suggestions for Authors

This manuscript presents an interesting study on the effectiveness of tumor control using FLASH compared to conventional (CONV) radiotherapy. The authors use the Py117 tumor cell line, inoculated in the left 4th mammary fat pad of a mouse, and evaluate treatment outcomes based on tumor eradication and growth delay. This research has the potential to contribute to the clinical translation of FLASH therapy.

However, there are concerns regarding the dosimetry description and the depth of result analysis. Below are some suggestions for the authors' consideration:

Major:

1) There are several inconsistencies in the descriptions of key parameters. For example: Table 1 SSD vs. line 244, Figure 1E vs. line 306, Table 2 dose rate vs. line 238, and Table 2 FLASH dose per pulse vs. intra-pulse dose rate. Additionally, the dose values related to the CONV pulse (1901, 4763) require clarification. The authors should carefully review these values for accuracy and consistency and consider revising Table 2 for improved clarity.

2) The definition of the dose reference position is unclear. Based on line 264, it appears that the dose is reported using film placed on top of the collimator, which may then be converted to an ion chamber reading inside a solid water phantom. However, neither of these directly corresponds to the dose in the tumor/mouse. Perhaps I missed something, but it seems problematic to assume that the same ion chamber reading ensures an identical dose at both 18.7 cm from the scattering foil and 76.1 cm away. Further clarification on this point would be helpful.

3) More details on the evaluation metrics and statistical analysis used in Figure 3 and Figure 4 are necessary. Statements such as “tumor control was equivalent” (line 415) and “without significant statistical difference” (line 399) are not acceptable. The authors should provide explicit statistical tests and confidence intervals to support these conclusions.

4) Why not shift the collimator so that the central uniform field can be utilized? Additionally, it is unclear whether the PDD in Figure 2F was measured in air or if a phantom was placed adjacent to the film to simulate tumor geometry within a mouse. Clarifying this would improve the interpretation of the data.

5) Line 387 states that a 3.75 mm treatment margin was associated with better eradication than a 2.5 mm margin, but this conclusion is not clearly supported by Figure 3. The results are displayed in a mixed manner for 20 Gy and 25 Gy, making it difficult to draw a definitive conclusion. Furthermore, for 30 Gy, experiments were halted due to severe tissue toxicity (line 401). In my opinion, if severe toxicity occurs, it suggests that the treatment parameters (either dose or field size) may not be appropriate for clinical application. Otherwise, one could argue that FLASH and CONV have the same effectiveness if the prescription were 100 Gy total body irradiation, where all subjects would succumb to treatment. The discussion should address this issue more carefully.

6) How are the treatment margin values defined? This is particularly unclear for large tumor cases. Additional explanation would improve understanding.

Minor:

1) line 242, differences?

2) line 248, probably consider descriptions with different symbols rather than with different colors.

3) it is difficult to differentiate between FLASH and CONV in Figure 2D and E due to the poor quality and similar symbols, although it is somewhat easy to differentiate X and Y.

4) line 317, FWHM differences?

5) missing arrows as described in the caption of Figure 4.

6) line 373, it gives me the impression that the matched beam geometry can reduce radiotoxicity. But I think the matched beam geometry is only related to providing matched dose profiles and has nothing to do with toxicity, whose reduction is done by reduced margin. Please consider rephrasing or correct me if I am wrong.

Author Response

We sincerely appreciate the reviewer for taking the time to evaluate our work. Your thoughtful insights have not only clarified key aspects of our study but have also raised crucial questions, ultimately enhancing the overall clarity of our data presentation.

Reviewer 2

This manuscript presents an interesting study on the effectiveness of tumor control using FLASH compared to conventional (CONV) radiotherapy. The authors use the Py117 tumor cell line, inoculated in the left 4th mammary fat pad of a mouse, and evaluate treatment outcomes based on tumor eradication and growth delay. This research has the potential to contribute to the clinical translation of FLASH therapy.

However, there are concerns regarding the dosimetry description and the depth of result analysis. Below are some suggestions for the authors' consideration:

Major:

1) There are several inconsistencies in the descriptions of key parameters. For example:

Response 1. We apologize for the poor reporting of our parameters and we thank the reviewer for the opportunity to correct these errors.

Table 1 SSD vs. line 244,

The table should report CONV SSD= 76.1 cm, as line 244 reports and Figure 1E shows. Table now reads:

Table 1. Experimental Irradiated Groups of Breast Cancer Orthotopic Tumor-Bearing Mice, Irradiated with Either FLASH or CONV Dose Rates in Two Irradiation Rounds with Different Abdominal Wall Treatment Fields.

Round 1

Mode

Tumor Volume

Dose

No. mice

Collimation

AWTFa

SSDb

Missed

[mm3]

[Gy]

 [cm2]

 [cm2]

[cm]

Pulses*

FLASH

20-40

20

4

2 × 2

0.75 × 2.00

18.7

1

FLASH

20-40

30

4

2 × 2

0.75 × 2.00

18.7

0

FLASH

250-800

30

6

2 × 2

0.75 × 2.00

18.7

0

CONV

20-40

20

4

2 × 2

0.75 × 2.00

76.1

-

CONV

20-40

30

4

2 × 2

0.75 × 2.00

76.1

-

CONV

250-800

30

6

2 × 2

0.75 × 2.00

76.1

-

Round 2

Mode

Tumor Volume

Dose

No. mice

Collimation

PTVa

SSDb

Missed

[mm3]

[Gy]

 [cm2]

 [cm2]

[cm]

Pulses*

FLASH

20-40

20

4

2 × 2

0.50 × 2.00

18.7

0

FLASH

20-40

25

8

2 × 2

0.50 × 2.00

18.7

1

FLASH

20-40

30

4

2 × 2

0.50 × 2.00

18.7

0

CONV

20-40

20

4

2 × 2

0.50 × 2.00

18.7

-

CONV

20-40

25

8

2 × 2

0.50 × 2.00

18.7

-

CONV

20-40

30

4

2 × 2

0.50 × 2.00

18.7

-

a Abdominal wall treatment field; treatment area within the mouse positioner frame (abdominal wall exposure).

b Source to surface distance.

* Animals irradiated with missed pulses were excluded from the analysis.

Figure 1E vs. line 306,

Previous line 306 is now corrected and reads as follows:

Line 319-321: In the first round of irradiations, the initial setup featuring varying beam geometries—76.1 cm for FLASH and 18.7 cm for CONV surface distance from the scattering foil, as shown in Figure 1E.

Table 2 dose rate vs. line 238, Table 2 FLASH dose per pulse vs. intra-pulse dose rate. Additionally, the dose values related to the CONV pulse (1901, 4763) require clarification. The authors should carefully review these values for accuracy and consistency and consider revising Table 2 for improved clarity.

CONV

During Round 1, the 16 MeV energy board was in clinical use; thus, gun intensity remained at clinical settings for CONV irradiation, resulting in an SSD of 76.1 cm (ref 45: Wang et al Med Phys 2024, doi:10.1002/mp.17432). A repetition rate of 4 (72 Hz) was utilized, and the delivered doses were as follows:

  • 20 Gy: 913 MU over 137.0 s (9860 pulses)
  • 30 Gy: 1370 MU over 205.5 s (14796 pulses)

In both cases, the average dose rate was 0.146 Gy/s, the dose per pulse was 2.03 × 10⁻³ Gy, and the intra-pulse dose rate was 541 Gy/s.

During Round 2, the 16 MeV energy board was decommissioned from clinical use, allowing matched geometry (SSD of 18.7 cm) to be achieved by reducing the gun intensity. Treatments were delivered using a repetition rate of 5 (90 Hz):

  • 20 Gy: 88 MU over 146.7 s (13200 pulses)
  • 25 Gy: 110 MU over 183.3 s (16500 pulses)
  • 30 Gy: 132 MU over 220.0 s (19800 pulses)

For these conditions, the average dose rate was 0.136 Gy/s, the dose per pulse was 1.52 × 10⁻³ Gy, and the intra-pulse dose rate was 404 Gy/s.

 FLASH

For FLASH irradiation (Rounds 1 and 2, SSD = 18.7 cm), the 20 MeV energy board was used at a repetition rate of 5 (90 Hz):

  • 20 Gy: 10 pulses at 2 Gy/pulse delivered in 0.1 s (effective dose rate: 200 Gy/s).
  • 30 Gy: 15 pulses at 2 Gy/pulse delivered in 0.156 s (effective dose rate: 192.86 Gy/s).
  • 25 Gy: 25 pulses at 1 Gy/pulse delivered in 0.267 s (effective dose rate: 93.75 Gy/s).

The intra-pulse dose rates were 5.33 × 10⁶ Gy/s for 2 Gy/pulse and 2.67 × 10⁶ Gy/s for 1 Gy/pulse.

Table 2 has been revised, and its format has been adjusted to match that of Table 1, improving clarity:

Table 1. Beam Parameters for Experimental Beam Delivery with FLASH or CONV

Round 1

Mode

Rxa

Pulses

DPPb

Pulse Rate

Dose Rate

Pulse Length

IPDRc

Energy

[Gy]

[Gy]

[Hz]

[Gy/s]

[s]

[Gy/s]

[MeV]

FLASH

20

10

2

90

200

3.75×10-6

5.33×105

16.60

FLASH

30

15

2

90

193

3.75×10-6

5.33×105

16.60

CONV

20

9860

2.03×10-3

72

0.146

3.75×10-6

541

15.73

CONV

30

14796

2.03×10-3

72

0.146

3.75×10-6

541

15.73

Round 2

Mode

Rx

Pulses

DPPa

Pulse Rate

Dose Rate

Pulse Length

IPDRb

Energy

[Gy]

[Gy]

[Hz]

[Gy/s]

[s]

[Gy/s]

[MeV]

FLASH

20

10

2

90

200

3.75×10-6

5.33×105

16.60

FLASH

25

25

1

90

94

3.75×10-6

2.67×105

16.60

FLASH

30

15

2

90

193

3.75×10-6

5.33×105

16.60

CONV

20

13200

1.52×10-3

90

0.136

3.75×10-6

404

15.73

CONV

25

16500

1.52×10-3

90

0.136

3.75×10-6

404

15.73

CONV

30

19800

1.52×10-3

90

0.136

3.75×10-6

404

15.73

a Rx: prescribed dose (target dose)

DPP: dose per pulse

c IDDR: intra-pulse dose rate.

2) The definition of the dose reference position is unclear. Based on line 264, it appears that the dose is reported using film placed on top of the collimator, which may then be converted to an ion chamber reading inside a solid water phantom. However, neither of these directly corresponds to the dose in the tumor/mouse. Perhaps I missed something, but it seems problematic to assume that the same ion chamber reading ensures an identical dose at both 18.7 cm from the scattering foil and 76.1 cm away. Further clarification on this point would be helpful.

Response 2. We thank the reviewer for highlighting this important point. The reference dose position was defined at the surface (top) of the collimator, measuring the tumor’s "entrance dose." Prior to the experiments, the ion chamber and films were employed to calibrate and verify agreement between FLASH and CONV doses. During each irradiation experiment, a dedicated film was placed at this reference position to measure the entrance dose delivered to each mouse (one film per mouse). Ion chamber measurements were not used independently to determine dose; instead, they were combined with experimental film data to reduce variability by correlating chamber charge with film-measured doses. We acknowledge that the measured entrance dose at this reference position may not directly reflect the exact dose within the tumor. However, especially for small tumors, we consider this a reliable approximation. For larger tumors, accuracy may diminish, which is why percentage depth dose (PDD) curves have been included to aid dose interpretation. Despite these limitations, we believe our approach provides a robust and consistent basis for comparing tumor responses within similarly sized groups.

We have added the following clarification to the Methods section, 2.7 Dosimetry:

Line 276-284: Absolute target doses were determined at the surface of the beam collimator using radiochromic film (EBT3 Galfchromic, Ashland Inc., Wayne, NJ; Figure 2A,B) and exit charge measurements at the Bremsstrahlung tail of the electron beam using an ion chamber (Farmer® Chamber, PTW Model TN30013, Boonton, NJ; Figure 1E). This reference point aims to represent the tumor’s entrance surface dose. Prior to experiment, films and chamber measurements were used to calibrate target doses of FLASH and CONV. During each mouse irradiation, one 2.4 × 5.1 cm piece of radiochromic film per mouse was placed under the stereotactic mouse positioner, on the top of the beam collimator, centered on the 2.0 × 2.0 cm irradiation field (Figure 2C).

3) More details on the evaluation metrics and statistical analysis used in Figure 3 and Figure 4 are necessary. Statements such as “tumor control was equivalent” (line 415) and “without significant statistical difference” (line 399) are not acceptable. The authors should provide explicit statistical tests and confidence intervals to support these conclusions.

Response 3. We appreciate the reviewer's feedback and acknowledge the need for a clearer description of our statistical analysis. In response, we have revised the manuscript to provide a more detailed explanation of our statistical approach, ensuring transparency and clarity. Given the non-parametric distribution of tumor volume data and the small sample size, confidence intervals around median differences were not directly computed. Instead, we provide detailed Mann–Whitney U test results, including exact p-values and mean rank differences at each time point comparing FLASH and CONV groups. The Mann–Whitney U test with multiple comparisons corrected using the Holm–Šidák method and statistical significance defined as α = 0.05, is appropriate for comparing distributions without assuming normality and provides a robust measure of group differences in this setting. These statistics are now explicitly reported in the Methods and Results sections, and detailed in Supplementary Tables 1 and 2.

We have added a paragraph in the Methods section.

Line 357-372: 2.10 Statistics

Tumor volumes were compared between FLASH and CONV irradiated groups at each post-irradiation time point using the Mann–Whitney U test, with multiple comparisons corrected using the Holm–Šidák method and statistical significance defined as α = 0.05. Due to the non-parametric distribution of tumor volume data and the small sample size, confidence intervals around median differences were not directly computed. Statistical analyses and graphical representation of data were performed using GraphPad Prism (Version 10.0, GraphPad Software, San Diego, CA, USA). The 20 Gy and 30 Gy groups were irradiated separately in Round 1 and Round 2 (n = 4 per group per round), with slight variations in delivered dose and treatment margins between rounds. To address these variations, analyses were initially stratified by irradiation round to verify consistency of results before data from both rounds were merged to evaluate overall treatment effects. The 25 Gy groups, irradiated only in Round 2, were analyzed independently. Detailed results of Mann–Whitney U tests, including exact p-values and mean rank differences for each time point comparing FLASH and CONV groups, are provided in Supplementary Tables 1 and 2.

Supplementary Table 1. P-values from tumor volume comparisons between FLASH and CONV irradiation groups.

P-values were derived from Mann–Whitney U tests performed at each indicated day post-irradiation, comparing tumor volumes between FLASH and CONV groups. Analyses are presented separately for Round 1 (20 Gy and 30 Gy), Round 2 (20, 25, and 30 Gy), and merged datasets from both rounds. Unless otherwise specified, all analyses correspond to small size tumors (20–40mm³). The 30 Gy (Lg) group represents the analysis of large size tumors (250-800 mm³). Statistical significance was set at α = 0.05.

dAY POST

Round 1

Round 1

Round 1

Round 2

Round 2

Round 2

Merged

Merged

RT

20Gy

30Gy

30Gy (Lg)

20Gy

25Gy

30Gy

20Gy

30Gy

0

0.885714

0.342857

0.818182

0.142857

0.804662

>0.999999

0.243978

0.243978

2

0.485714

0.771429

0.937229

0.714286

0.005905

0.685714

0.48345

0.48345

4

0.485714

>0.999999

0.937229

0.914286

0.895105

0.114286

0.409324

0.409324

6

0.971429

0.142857

0.818182

0.571429

0.82906

0.371429

0.773893

0.773893

10

0.657143

0.828571

0.484848

0.371429

0.577156

0.142857

0.253768

0.253768

12

0.257143

>0.999999

0.484848

>0.999999

0.82906

>0.999999

0.307071

0.307071

16

>0.999999

>0.999999

0.818182

0.314286

0.140637

>0.999999

0.706294

0.706294

23

>0.999999

>0.999999

0.588745

0.342857

0.043823

>0.999999

0.647552

0.647552

30

>0.999999

>0.999999

0.2

0.271062

>0.999999

0.386325

0.386325

33

>0.999999

>0.999999

0.7

0.062771

0.428571

0.5338

0.5338

39

0.914286

0.4

0.309524

0.828571

0.729021

0.729021

42

>0.999999

0.7

0.32684

0.828571

0.83683

0.83683

46

0.914286

0.8

0.132035

0.828571

0.848485

0.848485

Supplementary Table 2. Mean rank differences between FLASH and CONV groups at each post-irradiation day.

Mean rank differences (FLASH minus CONV) were calculated from Mann–Whitney U tests at each indicated time point post-irradiation, with positive values indicating higher tumor volumes (higher ranks) in the FLASH group and negative values indicating higher tumor volumes in the CONV group. Unless otherwise specified, all analyses correspond to small size tumors (20–40mm³). The 30 Gy (Lg) group represents the analysis of large size tumors (250-800 mm³).

dAY POST

Round 1

Round 1

Round 1

Round 2

Round 2

Round 2

Merged

Merged

RT

20Gy

30Gy

30Gy (Lg)

20Gy

25Gy

30Gy

20Gy

30Gy

0

0.5

2

-0.6667

-2.625

0.6696

0

-2.813

-2.813

2

-1.5

0.75

-0.3333

-0.875

-6.161

-1

-1.741

-1.741

4

1.5

0

0.3333

0.5833

0.4018

-3

2.009

2.009

6

-0.25

-3

-0.6667

1.167

0.5357

1.75

0.8036

0.8036

10

1

0.5

-1.667

1.458

1.339

2.75

2.679

2.679

12

2.25

-0.25

-1.667

0

-0.5357

0.5

2.411

2.411

16

1

0

-0.6667

-2.042

3.214

1

-0.8036

-0.8036

23

1

0

-1.333

-1.75

4.554

1

-1.071

-1.071

30

0.5

0

-2.333

2.625

1

-2.009

-2.009

33

0

0

-1

4

2

-1.429

-1.429

39

0.5

-1.667

2.333

0.5833

-0.8571

-0.8571

42

0

-1

2.167

0.5833

-0.5714

-0.5714

46

0.5

-0.8333

3.333

0.5833

0.4643

0.4643

We have added to the manuscript, Results section, the following clarifications: 

Line 392-404: No statistically significant differences were observed between FLASH- and CONV-irradiated tumors at any time point after irradiation for both the 20 Gy and 25 Gy groups (Supplementary Table 1). Specifically, for the 20 Gy group, the mean rank difference in tumor volumes between FLASH and CONV at day 46 was 0.464 mm³ (Mann–Whitney U test, p = 0.848), indicating no significant difference. Similarly, for the 25 Gy group, the mean rank difference was 3.333 mm³ (p = 0.132), also showing no statistical significance. Notably, tumors treated with 20 Gy had a wider treatment margin in Round 1 (3.75 mm) compared to Round 2 (2.5 mm), whereas tumors treated with 25 Gy were all treated in Round 2, consistently receiving a 2.5 mm treatment margin. The 20 Gy data are presented as a pooled dataset from both rounds; however, independent analyses were also conducted separately for Round 1 and Round 2, showing no statistically significant differences between FLASH and CONV at any time point post-irradiation (Supplementary Tables 1 and 2).

Line 420-427: No statistically significant differences were observed between 30Gy FLASH- and 30 Gy CONV-irradiated tumors at any time point after irradiation for both 3.75 mm treatment margin to tumor at Round 1 and or 2.5 mm for Round 2 (Supplementary Table 1). Specifically, for the 30 Gy group at Round 1 (Figure 1C), the mean rank difference in tumor volumes between FLASH and CONV at day 33 was 0 mm³ (Mann–Whitney U test, p > 0.999), indicating no significant difference. Similarly, for the 30 Gy group at Round 2 (Figure 3D) at day 46, the mean rank difference was 0.583 mm³ (p = 0.829), also showing no statistical significance.

Line 433-448: Figure 3. (A-D) Tumor measurements with calipers plotted as response curves of breast cancer orthotopic tumors of small volume (20-40 mm3) irradiated with 20, 25 and 30 Gy single fraction with either FLASH or CONV dose rates. (A) Tumor volumes from animals irradiated with 3.75 and 2.5 mm treatment margin to tumor combined (n=7 FLASH, n=8 CONV). FLASH group had one animal excluded due to a missed pulse. Tumors were controlled for the first 4 weeks and regressed thereafter, with no significant differences between groups at any time point post-irradiation (Mann–Whitney U test, all p > 0.05). (B) Tumors targeted with 25 Gy and 2.5 mm treatment to tumor margin (n=7 FLASH, n=8 CONV); one FLASH exclusion. Tumors remain controlled for 4 weeks and regressed thereafter, with no differences between groups (Mann–Whitney U test, all p > 0.05). (C) Tumor treated with 30 Gy and 3.75 mm treatment margin to tumor (n=4 per group) were controlled by day 30, but severe tissue toxicity led to study termination. No significant differences observed (all p > 0.05). (D) Tumors treated with 30 Gy and 2.5 mm treatment margin to tumor remained controlled for the first 4 weeks (all p > 0.05). By day 48, only one tumor per group showed regrowth (mean rank difference = 0.583 mm³; Mann–Whitney U test, p = 0.829). Overall, there was no significant difference between FLASH and CONV in tumor growth delay or eradication of small tumor volumes (20–40 mm³) with single fractions of 20, 25 and 30 Gy.

Line 456-467: In the initial week, unirradiated Py117 orthotopic tumors typically reach a volume of about 30 mm³, escalating to ~ 635 mm³ by the fourth week, and surpassing 1000 mm³ in the fifth week, as shown in Figure 4A. While smaller tumors display a consistent size range, larger ones exhibit greater variability in size. Larger tumors, with mean volumes of 493.05 ± 209.04 mm³ in the FLASH group and 497.71 ± 195.72 mm³ in the CONV group, exhibited similar regression within the first two weeks following treatment with a 30 Gy dose with either FLASH or CONV radiation, followed by a parallel pattern of regrowth. No statistically significant differences were observed between the two modalities at any time point post-irradiation (Supplementary Table 1). Specifically, by day 16, the mean rank difference in tumor volumes between FLASH and CONV was -0.667 mm³ (Mann–Whitney U test, p = 0.818), and by day 23, it was -1.333 mm³ (Mann–Whitney U test, p = 0.589), indicating no significant difference.

Line 484-488: Tumor volumes were suppressed for the first 2 weeks and regressed thereafter with no statistically significant differences observed between FLASH and CONV at any time point post-irradiation (Mann–Whitney U test, all p > 0.05). There was no significant difference between FLASH and CONV in tumor growth delay of large tumor volumes (250–800 mm³) with a single fraction of 30 Gy.

4) Why not shift the collimator so that the central uniform field can be utilized? Additionally, it is unclear whether the PDD in Figure 2F was measured in air or if a phantom was placed adjacent to the film to simulate tumor geometry within a mouse. Clarifying this would improve the interpretation of the data.

Response 4. Considering Figures 2D and 2E, the salmon-highlighted area represents the section of the stereotactic positioner that lies within the irradiation field in the X direction. The tumor is positioned at the right edge of this highlighted area, as shown in Figure 1A.

a) Effect of Collimator Movement:

Moving the collimator in either direction would not change the shape of the dose profiles within the irradiation field. This is because the profiles result directly from collimator wall scattering and source-to-field distance, both of which remain constant.

b) Effect of Moving the Stereotactic Positioner:

If the stereotactic positioner is shifted relative to the collimator to center the tumor within the irradiation field, the abdomen of the mouse will be extensively exposed, leading to severe gastrointestinal (GI) toxicity complications.

c) Field Homogeneity & Dose Distribution:

As seen at the right edge of the salmon highlight in Figure 2D, the tumor is positioned in an area where the dose profiles from the unmatched geometry remain highly comparable.

As detailed in Section 2.8 Beam Homogeneity and Dose Distribution (original manuscript lines 310-314):

A notable 14.5% difference in dose profiles at the field's perimeter prompted the incorporation of a wide abdominal wall treatment field of 7.5 × 20 mm2 within the mouse stereotactic positioner, yielding a minimal 0.9% dose variance at the positioner's trans-verse aperture, where the tumor protrudes (Table 1; Figure 1C.i and Figure 2D).

 The PDDs were not assessed in air. As stated in the Methods Section, 2.7 Dosimetry, at the end of the paragraph (original manuscript lines 280-283):

The percentage dose depth (PDD) distributions of the beam were assessed for both FLASH and CONV geometries using sagittal films, oriented parallel to the beam (Z direction), sandwiched between solid water (Figure 2F).

5) Line 387 states that a 3.75 mm treatment margin was associated with better eradication than a 2.5 mm margin, but this conclusion is not clearly supported by Figure 3. The results are displayed in a mixed manner for 20 Gy and 25 Gy, making it difficult to draw a definitive conclusion.

Response 5. We appreciate the reviewer’s concern about our initial claim that a 3.75 mm treatment margin provided better tumor eradication than a 2.5 mm margin. While a larger margin appeared to improve tumor control up to 30 days, its long-term efficacy remains unclear without extended follow-up. Accordingly, we have removed any definitive statements regarding the superiority of the 3.75 mm margin. We emphasize that this revision does not diminish the overall significance of our findings, which highlight comparable tumor control using FLASH and CONV in this orthotopic tumor-mouse model. We have revised the Abstract:

Line 37-45: (~3.75 or 2.5 mm treatment margin to tumor). Results: Both FLASH and CONV demonstrated comparable efficacy. Small tumors treated with 30 Gy and larger abdominal wall treatment fields appeared to have complete eradication at 30 days but also exhibited the highest skin toxicity, limiting follow-up and preventing confirmation of eradication. Smaller abdominal wall treatment fields reduced skin toxicity and allowed for extended follow-up which resulted in 75% tumor-free survival at 48 days. Larger tumors showed growth delay but no eradication. Conclusions: In this preclinical, non-metastatic orthotopic breast cancer model, FLASH and CONV demonstrated equivalent tumor control with single-fraction doses of 20, 25, or 30 Gy. Overall, 30 Gy achieved the highest eradication rate but also resulted in the most pronounced skin toxicity.

We have revised the Results 3.1:

Line 433-436: Among the 20, 25, and 30 Gy doses, 30 Gy achieved the highest tumor eradication rate but also produced the most pronounced skin toxicity. Specifically, administering 30 Gy with a 3.75 mm treatment margin caused greater abdominal wall skin exposure—thus limiting follow-up beyond 30 days—compared to using a 2.5 mm margin.

We also revised the Discussion:

Line 528-530: We also demonstrated for both FLASH and CONV radiotherapy comparable eradicate rates of small breast cancer nodules up to 48 days, with 30 Gy demonstrating the highest eradication rate.

We have revised the Conclusion:

Line 596-599: However, we also demonstrated for the first time, to our knowledge, that single dose FLASH radiotherapy has comparable eradication rate for small breast cancer foci equivalent to conventional dose rate radiotherapy in a syngeneic, orthotopic murine model of breast cancer.

Furthermore, for 30 Gy, experiments were halted due to severe tissue toxicity (line 401). In my opinion, if severe toxicity occurs, it suggests that the treatment parameters (either dose or field size) may not be appropriate for clinical application. Otherwise, one could argue that FLASH and CONV have the same effectiveness if the prescription were 100 Gy total body irradiation, where all subjects would succumb to treatment. The discussion should address this issue more carefully.

Response 5 cont.. We also recognize that skin toxicity at 30 Gy may raise questions about its clinical suitability. In our first round of experiments, mice receiving 30 Gy were euthanized once more than one mouse developed oozing ulcerations, preventing longer-term observation, as stated in the first paragraph of the 3.1 Results section (original manuscript lines 348-349):

Mice were euthanized at initial sign of ulceration without treatment. Autopsies were not performed on mice found dead.

Retrospectively, this limited our follow-up period and our knowledge of cause of death, though our primary aim was to evaluate whether our radiation methodology could show comparable tumor control. In order to reduce toxicity in the second round, we reduced the treatment margin, allowing a longer follow-up period. Moreover, clinical dosimetry is significantly more refined than our preclinical setup, and patients typically receive additional skin care, supporting the notion that 30 Gy can remain within a clinically acceptable dose range. We have added a Methods section clarifying the euthanasia criteria:

Line 178-182: 2.4  Euthanasia criteria

If any mouse developed ulceration in the tumor area, it was euthanized. Furthermore, if more than one mouse within the same dose group experienced ≥50% ulceration (with or without alopecia) in the irradiated field, the entire group was terminated without providing skin treatment.

Our decision to use the left 4th mammary fat pad was guided by prior work as described in the Discussion section (original manuscript lines 458-466):

Interestingly, while we noted skin toxicity with 30 Gy FLASH in this study, our group previously did not see lethal skin toxicity at 30 Gy when radiotherapy was applied to the hind limb [19]. We also failed to see skin toxicity in a model of unilateral, left chest wall irradiation with 6 months of follow-up (unpublished). For this study, we elected to use the 4th mammary fat pad in order to assess the anti-tumor effect of FLASH vs CONV with larger tumors in an orthotopic location: it would not have been feasible to grow larger tumors if we had used the 3rd mammary fat pad as this would have impaired the motility of mice and led to euthanasia of study animals before our tests could be completed.

We have added to the Discussion section the following suggestion:

Line 540-543: Therefore, considering the ideal placement of orthotopic tumors for investigating tumor eradication, the 3rd mammary fat pad, together with appropriate skin treatments for acute radiation toxicity, would facilitate reliable long-term monitoring by mitigating the limitations posed by skin toxicity.

6) How are the treatment margin values defined? This is particularly unclear for large tumor cases. Additional explanation would improve understanding.

Response 6. We acknowledge the lack of description and have now clarified our definition of the treatment margin as the minimum distance between the tumor epicenter and the edge of the radiation treatment field.

To improve clarity and avoid confusion, we have revised the Methods section. Specifically, we relocated part of the description of the minimum treatment margin to tumor from Section 2.2 Orthotopic Mouse Model to Section 2.6 Irradiation Study Design, where it is more appropriately placed. Additionally, we replaced the term planning treatment volume (PTV) in Table 1 with abdominal wall treatment field (AWTF), as noted in our response to Major Comment 1, and updated the abbreviation list. The revised paragraph (b) in Section 2.6 Irradiation Study Design now reads:

Line 256-263: b) In Round 1, a 7.5 × 20 mm² abdominal wall treatment field (AWTF; Table 1) was used, providing an approximate minimum treatment margin to tumor of 3.75 mm, which is defined as the minimum distance between the tumor epicenter and the edge of the radiation treatment field (Figure 1C.i). In Round 2, a 5 × 20 mm² abdominal wall treatment field was employed (Figure 1C.ii; Table 1), yielding an approximate minimum treatment margin to tumor of 2.5 mm. For simplicity, the minimum treatment margin to tumor distance will henceforth be referred to as “treatment margin” throughout the manuscript and figures.

Minor:

  • line 242, differences?

Response 1. Corrected

2) line 248, probably consider descriptions with different symbols rather than with different colors.

Response 2. Thank you for the suggestion. We have implemented this change by using the CONV* notation for the matched geometry instead of color differentiation. Corresponding adjustments have been made to the figure, figure legend, and relevant sections of the manuscript.

Figure 1. In vivo breast cancer orthotopic tumor irradiations with FLASH or CONV dose rates. (A) Breast cancer orthotopic tumor-bearing mouse with approximately 30 mm³ tumor at the 4th mammary fat pad (top; black dashed circle and arrow). An anesthetized mouse placed inside the mouse positioner frame with the tumor centered at the bottom of the lateral opening and immobilized with paper tissue (bottom). (B) 3D computer-aided design (CAD) files illustrating the positioning of the mouse positioner on the collimator, the positioning of the radiochromic film during irradiation. (C) Top view of the collimator featuring the mouse positioner in two treatment margins to tumor: (i) Round 1, positioned to expose a 7.5 × 20.0 mm² area of the abdominal wall tissue, 3.75 mm margin or (ii) Round 2, positioned to expose a 5.0 × 20.0 mm² area of the ab-dominal wall tissue, 2.5 mm margin. (D) 3D CAD file of the 2.0 × 2.0 cm² collimator presented with a lateral cross-section, illustrating mouse radiation shielding. The collimator is filled with a 3-cm-layer of aluminum oxide to stop the electrons and a 1-cm-layer of tungsten spheres (2.0 mm diameter) to absorb Bremsstrahlung radiation and efficiently shield the rest of the animal’s body (Bremsstrahlung radiation leakage ~0.2%). (E) FLASH and CONV beam geometries. For FLASH irradiations, the collimator and mouse positioner frame are placed inside the treatment head and the beam entrance surface of the mouse is 18.7 cm from the scattering foil. For CONV irradiations, at Round 1, the beam entrance surface of the mouse was 76.1 cm (unmatched geometries) from the scattering foil, and at Round 2 at 18.7 cm (CONV* notation matching geometry with FLASH). The pulses delivered are monitored using an ion chamber, measuring the Bremsstrahlung tail at 11.0 cm solid water depth, and is located 147.5 cm from the scattering foil.

Line 249-253: Radiotherapy treatments were administered in two sequential distinct experiments, Round 1 and 2, with some key differences between them: a) in Round 1 the source-to-surface distance (SSD; scattering foil to mouse surface) for FLASH beam geometry was 18.7 cm compared to 76.1 cm SSD for the CONV beam geometry (Figure 1E, CONV vs FLASH notation), as described previously [46,47]. On Round 2, due to improvements in our configuration to the beam geometry the mismatch was resolved and all experiments thereafter implemented same geometry between modalities (Figure 1E, CONV* vs FLASH notation).

3) it is difficult to differentiate between FLASH and CONV in Figure 2D and E due to the poor quality and similar symbols, although it is somewhat easy to differentiate X and Y.

Response 3. We have updated the figures by using "X" symbols for the X profiles and inverted triangles for the Y profiles. Additionally, we have increased the border thickness and maintained the color coding corresponding to the different SSDs to ensure consistency with Figure 2F. We hope these changes improve readability and clarity for Figures 2D and 2E.

4) line 317, FWHM differences?

Response 4. Corrected

5) missing arrows as described in the caption of Figure 4.

Response 5. Corrected

6) line 373, it gives me the impression that the matched beam geometry can reduce radiotoxicity. But I think the matched beam geometry is only related to providing matched dose profiles and has nothing to do with toxicity, whose reduction is done by reduced margin. Please consider rephrasing or correct me if I am wrong.

Response 6. Thank you for your comment. You are correct, and we appreciate the opportunity to clarify this point. The matched beam geometry is intended to provide consistent dose profiles rather than directly affecting radiotoxicity. The text has been revised for clarity.

Lines 413-415: In Round 2 with matched beam geometry, in order to minimize radiotoxicity a smaller 5 × 20 mm2 abdominal wall treatment field (2.5 mm margin) was pursued (Table 1).

Round 2

Reviewer 1 Report

Comments and Suggestions for Authors

The revised version of the Manuscript is much improved. Although iam not still considering this study as something absolutely clear for or against FLASH, the authors have responded well and acknowledge the limitations. 

Reviewer 2 Report

Comments and Suggestions for Authors

Authors have addressed all of my questions.